# Change in the Physiological Aspects of Soybean Caused by Infestation by *Bemisia tabaci* MEAM1

Luciana B. Silva [1,*], Lucas C. Almeida [2], Maria C. F. e Silva [1], Ramilos R. de Brito [1], Rafael de S. Miranda [1],
Raimundo H. F. Rodrigues [3], Carlos M. P. dos Santos [1], Gilvana da S. Ribeiro [1], João V. S. Morais [1],
Alisson F. T. da Silva [1], Hernesise Mayard [1], Thayline Rodrigues de Oliveira [1],
Vânia Maria Gomes da Costa Lima [1], Lucia da Silva Fontes [1], Neurandir Sobrinho da Rocha [1],
Julian de Jesus Lacerda [1] and Bruno Ettore Pavan [4]

1   Graduate Program in Agricultural Sciences, Federal University of Piauí, Bom Jesus 64900-000, Brazil;
    mariacarolinafarias@outlook.com (M.C.F.e.S.); ramilos@ufpi.edu.br (R.R.d.B.);
    rsmiranda@ufpi.edu.br (R.d.S.M.); magnopiaui@ufpi.edu.br (C.M.P.d.S.);
    gilvanasribeiro@gmail.com (G.d.S.R.); joao.morais@ufpi.edu.br (J.V.S.M.); alisson@ufpi.edu.br (A.F.T.d.S.);
    hernesise.mayard@yahoo.com (H.M.); thaylinerodrigues@ufpi.edu.br (T.R.d.O.);
    vaniamgdcl@gmail.com (V.M.G.d.C.L.); lsfonte@ufpi.edu.br (L.d.S.F.);
    sobrinhoneurandi@gmail.com (N.S.d.R.); julianj@ufpi.edu.br (J.d.J.L.)
2   Department of Agronomy, Federal University of Piauí, Bom Jesus 64900-000, Brazil;
    lucascostared@hotmail.com
3   Department of Agronomy-Phytotechnics, Federal University of Ceará, Fortaleza 60020-181, Brazil;
    raimundoagro117@gmail.com
4   Department of Plant Science, Food Technology and Partner Economy, Júlio de Mesquita Filho Paulista State
    University, Ilha Solteira 15385-000, Brazil; be.pavan@unesp.br
*   Correspondence: lubarbosabio@ufpi.edu.br

**Abstract:** Whiteflies cause significant crop losses through direct sap feeding, inducing plant physiological disorders and promoting the growth of sooty mold. Moreover, whiteflies can indirectly harm plants by transmitting plant viruses, particularly begomoviruses and criniviruses, resulting in severe viral disease epidemics. This study aimed to evaluate the physiological characteristics of susceptible and resistant soybean cultivars to *B. tabaci*. The experiments were conducted in a greenhouse. Eleven soybean cultivars were selected and infested with 100 adults of *B. tabaci* at the V3 stage. The evaluation of photosynthetic parameters, such as photosynthetic rate, leaf transpiration, stomatal conductance, and internal $CO_2$ concentration, revealed that *B. tabaci* infestation influenced gas exchange in soybean plants. The photosynthetic rate was higher in cultivars AS3810 and M8349 during the V6 stage. Infestations caused alterations in photosynthetic parameters, suggesting increased energy demand to maintain photosynthetic activity. However, the response to infestation varied among the different cultivars, indicating varying levels of resistance and tolerance to the whitefly's damage. Furthermore, the infestation had a more notable impact during the vegetative phenological stage. In summary, infestation by *B. tabaci* has a discernible impact on the physiology of soybean plants, resulting in alterations in gas exchange parameters and water use efficiency. The reaction to infestation exhibited variations among different soybean cultivars, indicating potential differences in resistance to the pest. This study underscores the significance of assessing the physiological consequences of whitefly infestations on soybean crops.

**Keywords:** whitefly; photosynthetic parameters; gas exchange; insect pest; plants resistance; *Glycine max*

## 1. Introduction

*Bemisia tabaci* Gennadius (Hemiptera: Aleyrodidae) is among the most economically important insect pests of various vegetable crops in Brazil. This insect is considered a complex of at least 40 morphologically indistinguishable cryptic species. *Bemisia tabaci* Middle East–Asia Minor 1 (MEAM1) was initially introduced in Brazil around 1990, and

has since rapidly spread. Is a devastating cosmopolitan sap-sucking insect that poses a significant threat to various economically important crops, including soybean (*Glycine max* L.). As the second-most widespread and economically important arthropod pest globally, *B. tabaci* has been responsible for substantial yield losses in soybean crops [1–4]. The economic impact of this pest reaches hundreds of millions of US dollars annually across diverse agricultural production systems worldwide [3–8].

The direct and indirect plant damage caused by *B. tabaci* has led to substantial economic losses in soybean crops. *Bemisia tabaci* outbreaks on vegetables in Brazil resulted in significant economic losses of USD 132.3 and 161.2 million in 2016 and 2017, respectively [9–14].

Studies have revealed that whitefly infestations can deplete the energy reserves of plants, diminish primary production, and exert direct phytotoxic effects [15]. Whitefly infestations have been shown to reduce chlorophyll levels in various plant species, including tomato (*Solanum lycopersicum*), zucchini (*Cucurbita pepo*), eggplant (*Solanum melongena*), and soybean (*G. max*) [15–18]. This damage extends to the photosynthetic apparatus, particularly PSII and PSI, due to decreased stability in the oxygen-evolving complex and reaction centers of PSII, as well as a decline in electron transport [12]. Whitefly-infested tomato plants also display reduced parameters such as liquid photosynthesis, stomatal conductance, apparent carboxylation efficiency, and maximum efficiency of PSII [13].

Considering the substantial impact of *B. tabaci* MEAM1 on soybean crops, this study aimed to evaluate the physiological traits of both susceptible and resistant soybean cultivars to *B. tabaci*. Gaining insight into the specific physiological changes induced by the infestation is crucial for understanding the mechanisms of damage and devising potential strategies to effectively manage this destructive pest in soybean cultivation.

## 2. Materials and Methods

### 2.1. Rearing and Maintaining the Whitefly Population

The experiments were carried out at the Federal University of Piauí (UFPI-CPCE) (9°05004.4″ S, 44°19037.5″ W, 270 m). Whitefly nymphs were collected 3 months before the start of the experiments in tomato fields (9°01030.9″ S, 44°23021.2″ W, 353 m) under insecticide-free growing conditions. The insects were then brought to a laboratory to develop a rearing population. *B. tabaci* was housed in cages in the greenhouse. To prevent insects from escaping, PVC pipe construction cages (0.5 m × 0.5 m × 0.8 m) were coated with white voile fabric. The greenhouse had a shaded ceiling made of cloth and a transparent canvas that was closed laterally with an antiaphid screen (50 mesh). Leaf cabbage (*Brassica oleracea* L. var. sabellica, Brassicaceae) was used for whitefly breeding. It was grown in 5-liter pots and monitored daily to eliminate other insects. The cabbage plants were grown on a substrate consisting of soil, washed sand, and bovine manure (1:1:1), fertilized as recommended based on soil analysis.

### 2.2. Identification of Biotype B, Middle East–Asia Minor 1 (MEAM1)

For identification of the biotype, pumpkin plants were placed inside cages with the whiteflies and checked for the presence of the characteristic symptom of *B. tabaci* MEAM1 infestation, when the leaves become silvery. Silvery leaves are a typical response to the physiological disturbance caused by feeding of the insect on this culture. For confirmation, adults were sent to the entomology sector at Embrapa Arroz e Feijão (Santo Antônio de Goias, GO, Brazil), where molecular characterization was performed, and it was verified that the biotype of the whitefly used in the study was Middle East–Asia Minor 1 (MEAM1).

### 2.3. Soybean Cultivars

The eleven soybean cultivars were chosen based on prior research [19,20], which demonstrated their adaptability to the northeastern Brazilian environment and their ability to yield well. These cultivars can be considered valuable sources of resistance against *B. tabaci* MEAM1 for breeding programs aimed at developing resistant soybean cultivars [19,20] (Table 1).

**Table 1.** Soybean cultivars used in the experiments.

| Nº | Cultivar | Resistance History |
|---|---|---|
| 1 | AS 3810 IPRO[1] | Resistance—Antibiosis [20] |
| 2 | BONUS IPRO[2] | Susceptible [19,20] |
| 3 | BRS 9280RR | Resistance—Antixenosis and Antibiosis [19,20] |
| 4 | FTR 3190 IPRO | -- |
| 5 | M 8349 IPRO | -- |
| 6 | M 8644 IPRO | Antixenosis [19] |
| 7 | M 8808 IPRO | Resistance—Antixenosis and Antibiosis [19,20] |
| 8 | BRASMAX EXTREMA IPRO® | -- |
| 9 | BRASMAX DOMÍNIO IPRO® | -- |
| 10 | GSC F07 BT | -- |
| 11 | BRS 8383 IPRO | Resistance—Antibiosis [20] |

*2.4. Experimental Procedure*

The indicators for the chemical characterization of the samples, necessary for the recommendation of liming and fertilization in soybean crops, were obtained through the interpretation of soil analysis. Soybean plants were cultivated in pots containing 10 kg of substrate, composed of soil and cattle manure in a 3:1 ratio. In each pot, five seeds were planted, treated with insecticide and fungicide (Belure® + Vitavax®-Thiram 200 SC), and inoculated with bacteria of the genus Rhizobium.

Thinning was carried out after seed emergence, leaving only one plant per pot. Irrigation was performed according to the water requirements of the plants, with special attention given to irrigating at the base of the plant rather than wetting the leaf area to prevent the growth of phytopathogens. Six plants of each cultivar were used, where three were infested with whitefly and three were left free of infestation, resulting in a total of 66 experimental units.

When the soybean reached the V3 stage, 100 adults of *B. tabaci* were captured from the rearing cabbage plants, placed in glass test tubes, and released at the base of each soybean plant, allowing the whitefly to have a chance to choose the plant for infestation.

Two evaluations were carried out for each physiological parameter, with 6 readings per treatment. The photosynthetic rate (A $\mu$mol m$^{-2}$ s$^{-1}$), leaf transpiration (E mmol m$^{-2}$ s$^{-1}$), stomatal conductance (gs mol m$^{-2}$ s$^{-1}$), and internal $CO_2$ concentration (Ci $\mu$mol mol$^{-1}$) were measured using an infrared gas analyzer (IRGA, Portable Gas Exchange Fluorescence System® GFS-3000, Walz, Effeltrich, Germany) coupled with artificial light using blue and red light-emitting diodes (LEDs) with an intensity of 1200 $\mu$mol m$^{-2}$ s$^{-1}$. One reading per plant was performed in the morning (7:00 to 10:00 am) to determine gas exchange, using a fully expanded middle third leaf.

From the gas exchange data, the following relationships were calculated: instantaneous water use efficiency (EUA = A/E $\mu$mol $CO_2$/mmol$^{-1}$ $H_2O$), intrinsic water use efficiency (EIUA = A/gs $\mu$mol $CO_2$/mmol$^{-1}$ $H_2O$), and instantaneous carboxylation efficiency (A/Ci $\mu$mol m$^{-2}$ s$^{-1}$/$\mu$mol mol$^{-1}$). The assessments were conducted over four distinct periods. The initial and second readings took place during the V3 stage, one and two days post-infestation, respectively. The third evaluation occurred when the plants reached the V6 stage after fifteen days, and the final assessment was conducted at the R1 stage, twenty days following infestation.

*2.5. Statistical Analysis*

The physiological data were subjected to a triple factorial analysis of variance, involving 11 cultivars and five evaluation periods. As a triple interaction was detected for all the evaluated parameters, an analysis of canonical variables was conducted, considering the seven physiological parameters resulting from the interaction between the 11 cultivars and five evaluation periods. The statistical procedures were performed using GENES software 1 [21].

## 3. Results

According to the analysis of variance, significant interactions were observed in the photosynthetic parameters of soybean plants infested with *B. tabaci* (Table 2). However, in the interaction between the factors, whitefly infestation versus evaluation period, there was no significant effect ($p < 0.05$) on the photosynthetic rate (A), leaf transpiration (E), stomatal conductance (gs), and internal $CO_2$ concentration (Ci). Conversely, in the evaluations for photosynthetic rates, leaf transpiration, and gas exchanges, significant effects were observed in plants infested by *B. tabaci*, in the interaction between soybean cultivars and infestation (Table 2).

**Table 2.** Photosynthetic response, leaf transpiration (E), stomatal conductance (gs), photosynthetic rate (A), internal $CO_2$ concentration (Ci), instantaneous carboxylation efficiency (A/Ci), instantaneous water use efficiency (EUA), and intrinsic water use efficiency (EIUA) of soybean leaves subjected to injury from whitefly.

| Source of Variation | df | Medium Squares | | | | | | |
|---|---|---|---|---|---|---|---|---|
| | | E | gs | A | Ci | A/Ci | EUA | EIUA |
| Cultivars (C) | 10 | 2.476 ** | 58,472 ** | 22.34 ** | 21,421 ** | 0.033 ** | 2.18 ** | 0.017 ** |
| Infestation (I) | 1 | 167.8 ** | 449,097 ** | 98.45 ** | 53,924 ** | 0.037 ** | 218.2 ** | 0.076 ** |
| EP (E) | 3 | 8.06 ** | 142,370 ** | 129.9 ** | 103,422 ** | 0.114 ** | 11.62 ** | 0.066 ** |
| C × I | 10 | 4.38 ** | 8836 ** | 23.64 ** | 5738 ** | 0.031 ** | 5.20 ** | 0.003 ** |
| C × E | 30 | 3.85 ** | 28,115 ** | 19.85 ** | 17,693 ** | 0.030 ** | 7.89 ** | 0.009 ** |
| I × E | 3 | 1.18 ns | 976.1 ns | 4.86 ns | 267.7 ns | 0.017 * | 2.61 ** | 0.004 ** |
| C × I × E | 30 | 3.10 ** | 895.6 ns | 15.2 * | 5207 ** | 0.015 ** | 4.29 ** | 0.002 ** |
| Error | 176 | 0.70 | 3389 | 9.22 | 2096 | 0.005 | 0.64 | 0.0003 |
| Mean | | 3.99 | 210.3 | 16.50 | 214.6 | 0.106 | 4.63 | 0.102 |
| CV% | | 20.96 | 27.68 | 18.41 | 21.33 | 65.5 | 17.36 | 17.33 |

*, **, ns significant to 5, 1%, and not significant, respectively by the F test; EP, evaluation period. A: photosynthetic rate ($\mu$mol m$^{-2}$ s$^{-1}$), E: leaf transpiration (mmol m$^{-2}$ s$^{-1}$), gs: stomatal conductance (mol m$^{-2}$ s$^{-1}$), Ci: internal $CO_2$ concentration ($\mu$mol mol$^{-1}$), EUA: instantaneous water use efficiency (A/E $\mu$mol $CO_2$/mmol$^{-1}$ $H_2O$), EIUA: intrinsic water use efficiency (A/*gs* $\mu$mol $CO_2$/mmol$^{-1}$ $H_2O$), and A/Ci: instantaneous carboxylation efficiency ($\mu$mol m$^{-2}$ s$^{-1}$/$\mu$mol mol$^{-1}$).

The evaluation of photosynthesis (A) indicates that the photosynthetic rate was affected in cultivars under infestation. Notably, this impact was statistically significant for only three cultivars: AS 3810 IPRO, M8808, and M8349. Among these, AS3810 and M8808 exhibited resistance to *B. tabaci*, attributed to antixenosis and antibiosis. The V6 phenological stage emerged as a photosynthetically relevant phase for these cultivars. During this stage, the photosynthetic rate (A $\mu$mol m$^{-2}$ s$^{-1}$) was higher in AS3810 and M8349 among the infested plants, whereas in the case of M8808, it was lower (Figure 1).

Concerning the transpiration rate (E) parameter, significant differences were observed during the evaluation periods V3-2 and V6, corresponding to 48 h and fifteen days post-infestation with *B. tabaci*. The comparison between infested and non-infested plants revealed statistically significant variations (Table 2 and Figure 2). Notably, for the cultivars BRS9280 and DOMÍNIO, an elevation in the transpiration rate was noted at the R1 phenological stage in the non-infested plants.

In Figure 3, it can be observed that the stomatal conductance in soybean cultivars under *B. tabaci* infestation increased significantly. Specifically, in the vegetative stage, there was a significant increase in stomatal conductance in the cultivars EXTREMA, M8349, M8644, AS3810, M8644, and M8808. Additionally, in the reproductive stage, the cultivars DOMINIO, EXTREMA, and BRS9280 also showed a significant increase in stomatal conductance (Figure 3).

**A**

| CULTIVAR | STAGE | V3-1 | V3-2 | V6 | R1 |
|---|---|---|---|---|---|
| AS3810 | Non-infested | 1.00 | 1.00 | 1.00 | 1.00 |
| | Infested | 1.06 | 1.06 | 1.32 * | 1.12 |
| BONUS | Non-infested | 1.00 | 1.00 | 1.00 | 1.00 |
| | Infested | 1.24 | 0.73 | 1.38 | 1.22 |
| BRS8383 | Non-infested | 1.00 | 1.00 | 1.00 | 1.00 |
| | Infested | 1.03 | 0.95 | 1.02 | 1.05 |
| BRS9280 | Non-infested | 1.00 | 1.00 | 1.00 | 1.00 |
| | Infested | 1.34 | 1.34 | 0.88 | 1.27 |
| DIMÍNIO | Non-infested | 1.00 | 1.00 | 1.00 | 1.00 |
| | Infested | 0.97 | 0.85 | 0.85 | 1.60 |
| EXTREMA | Non-infested | 1.00 | 1.00 | 1.00 | 1.00 |
| | Infested | 1.25 | 1.25 | 1.24 | 1.22 |
| FTR3190 | Non-infested | 1.00 | 1.00 | 1.00 | 1.00 |
| | Infested | 0.96 | 0.91 | 0.90 | 0.70 |
| GCS | Non-infested | 1.00 | 1.00 | 1.00 | 1.00 |
| | Infested | 1.15 | 1.41 | 1.05 | 1.27 |
| M8349 | Non-infested | 1.00 | 1.00 | 1.00 | 1.00 |
| | Infested | 1.32 | 1.32 | 1.45 * | 0.79 |
| M8644 | Non-infested | 1.00 | 1.00 | 1.00 | 1.00 |
| | Infested | 0.75 | 0.75 | 1.23 | 0.84 |
| M8808 | Non-infested | 1.00 | 1.00 | 1.00 | 1.00 |
| | Infested | 1.16 | 1.08 | 0.94 * | 1.24 |

**Figure 1.** Heat map showing the photosynthetic $CO_2$ assimilation (A) of soybean cultivars in non-infested and infested plants. * indicate statistically significant differences; $p < 0.05$. The analyzed cultivars were AS 3810 IPRO1; BONUS IPRO2; BRS 9280RR, FTR 3190 IPRO, M 8349 IPRO; M 8644 IPRO; M 8808 IPRO; BRASMAX EXTREMA IPRO®; BRASMAX DOMÍNIO; PRO®; GSC F07 BT; and BRS 8383 IPRO. The data represent the change in A ($\mu$ mol $CO_2$) of infested plants at phenological stages V3-1; V3-2; V6; and R1 compared to their respective control conditions, non-infested plants. In the heat map, blue and orange represent up-regulated and down-regulated genes, respectively.

In the AS3810 cultivar, during the evaluation period V6, all of the photosynthetic parameters were significantly altered when comparing infested and non-infested plants, with higher activity observed in the infested plants (Figures 1–3). The blooming phase (R1) was identified as the phenological stage with the most significant changes in activity when subjected to infestation, based on the examination of instantaneous carboxylation efficiency (A/Ci), water use efficiency (WUE), and instantaneous water use efficiency (IWUE) (Supplementary Materials).

Concerning the internal concentration of $CO_2$ (Ci) as a photosynthetic parameter in the M8808 cultivar, elevated values were consistently observed in infested plants across all evaluation periods, emphasizing the enduring impact of *B. tabaci* infestation during various growth stages. Additionally, in cultivars Bonus, Domínio, FTR 3190, M8349, M8644, and M8808, the significant difference in Ci values was specifically noted during the evaluation period R1, highlighting a distinct response to infestation as the plants entered the reproductive stage (refer to Supplementary Materials for detailed data).

| CULTIVAR | STAGE | V3-1 | V3-2 | V6 | R1 |
|---|---|---|---|---|---|
| AS3810 | Non-infested | 1.00 | 1.00 | 1.00 | 1.00 |
| | Infested | 1.67 | 3.84 * | 2.38 * | 1.46 |
| BONUS | Non-infested | 1.00 | 1.00 | 1.00 | 1.00 |
| | Infested | 1.17 | 0.72 | 2.02 * | 1.96 |
| BRS8383 | Non-infested | 1.00 | 1.00 | 1.00 | 1.00 |
| | Infested | 1.48 | 1.30 * | 1.17 | 1.60 |
| BRS9280 | Non-infested | 1.00 | 1.00 | 1.00 | 1.00 |
| | Infested | 1.28 | 1.28 | 0.46 * | 2.10 * |
| DIMÍNIO | Non-infested | 1.00 | 1.00 | 1.00 | 1.00 |
| | Infested | 1.42 | 2.92 * | 0.96 | 3.50 * |
| EXTREMA | Non-infested | 1.00 | 1.00 | 1.00 | 1.00 |
| | Infested | 1.94 * | 1.94 * | 3.01 * | 1.59 * |
| FTR3190 | Non-infested | 1.00 | 1.00 | 1.00 | 1.00 |
| | Infested | 1.17 | 1.87 * | 1.17 | 0.97 |
| GCS | Non-infested | 1.00 | 1.00 | 1.00 | 1.00 |
| | Infested | 1.53 | 1.89 | 1.09 * | 2.00 |
| M8349 | Non-infested | 1.00 | 1.00 | 1.00 | 1.00 |
| | Infested | 1.89 * | 1.89 * | 2.24 * | 1.27 |
| M8644 | Non-infested | 1.00 | 1.00 | 1.00 | 1.00 |
| | Infested | 1.17 | 1.17 | 2.18 * | 1.06 |
| M8808 | Non-infested | 1.00 | 1.00 | 1.00 | 1.00 |
| | Infested | 1.94 * | 2.00 * | 1.57 * | 1.24 |

**Figure 2.** Heat map showing the transpiration (E) of soybean cultivars in non-infested and infested plants. * indicate statistically significant differences; $p < 0.05$. The analyzed cultivars were AS 3810 IPRO1; BONUS IPRO2; BRS 9280RR, FTR 3190 IPRO, M 8349 IPRO; M 8644 IPRO; M 8808 IPRO; BRASMAX EXTREMA IPRO®; BRASMAX DOMÍNIO; PRO®; GSC F07 BT; and BRS 8383 IPRO. The data represent the change in E (mmol $H_2O$ m$^{-2}$ S$^{-1}$) of infested plants at phenological stages V3-1; V3-2; V6; and R1 compared to their respective control conditions, non-infested plants. In the heat map, blue and orange represent up-regulated and down-regulated genes, respectively.

In terms of the instantaneous rate (USA) and intrinsic rate of water use (EIUA) parameters, the disparities between infested and non-infested cultivars were most evident during the vegetative phenological stage, with higher values recorded in the non-infested plants. However, in cultivars BRS 8383, BRS 9280, Bonus, Domínio, FTR 3190, and M8349, the significant difference emerged during the evaluation period R1, suggesting a shift in the physiological response to infestation as the plants progressed into the reproductive stage (refer to Supplementary Materials for detailed data).

In examining the efficiency of carboxylation (A/Ci), significant differences were identified during the evaluation period R1 for the cultivars BRS 9280, Domínio, FTR 3190, M8349, M8644, and M8808. Contrarily, the cultivar M8644 exhibited significant differences during the vegetative phenological stage (periods V3-1, V3-2, and V6). AS 3810, on the other hand, displayed significant differences solely in the period V6 (refer to Supplementary Materials for detailed data). These findings emphasize the nuanced and cultivar-specific responses to *B. tabaci* infestation across different physiological parameters and growth stages.

| CULTIVAR | STAGE | V3-1 | V3-2 | V6 | R1 |
|---|---|---|---|---|---|
| AS3810 | Non-infested | 1.00 | 1.00 | 1.00 | 1.00 |
| | Infested | 1.14 | 2.05 | 1.28 * | 1.61 |
| BONUS | Non-infested | 1.00 | 1.00 | 1.00 | 1.00 |
| | Infested | 1.52 | 0.71 | 2.25 | 2.02 |
| BRS8383 | Non-infested | 1.00 | 1.00 | 1.00 | 1.00 |
| | Infested | 1.49 | 1.17 | 1.36 | 1.69 |
| BRS9280 | Non-infested | 1.00 | 1.00 | 1.00 | 1.00 |
| | Infested | 1.68 | 1.68 | 0.59 * | 2.27 * |
| DIMÍNIO | Non-infested | 1.00 | 1.00 | 1.00 | 1.00 |
| | Infested | 0.88 | 1.68 | 1.21 | 4.72 * |
| EXTREMA | Non-infested | 1.00 | 1.00 | 1.00 | 1.00 |
| | Infested | 2.53* | 2.53* | 2.40 * | 1.85 * |
| FTR3190 | Non-infested | 1.00 | 1.00 | 1.00 | 1.00 |
| | Infested | 0.85 | 1.14 | 1.16 | 1.09 |
| GCS | Non-infested | 1.00 | 1.00 | 1.00 | 1.00 |
| | Infested | 1.19 | 1.44 | 1.08 * | 2.53 |
| M8349 | Non-infested | 1.00 | 1.00 | 1.00 | 1.00 |
| | Infested | 2.27 * | 2.27 * | 1.99 * | 1.27 |
| M8644 | Non-infested | 1.00 | 1.00 | 1.00 | 1.00 |
| | Infested | 1.25 | 1.25 | 2.05 * | 1.21 |
| M8808 | Non-infested | 1.00 | 1.00 | 1.00 | 1.00 |
| | Infested | 1.70 * | 1.41 | 1.61 * | 0.82 |

**Figure 3.** Heat map showing the stomatal conductance to water vapor (Gs) of soybean cultivars in non-infested and infested plants. * indicate statistically significant differences; $p < 0.05$. The analyzed cultivars were AS 3810 IPRO1; BONUS IPRO2; BRS 9280RR, FTR 3190 IPRO, M 8349 IPRO; M 8644 IPRO; M 8808 IPRO; BRASMAX EXTREMA IPRO®; BRASMAX DOMÍNIO; PRO®; GSC F07 BT; and BRS 8383 IPRO. The data represent the change in E (mmol $H_2O$ $m^{-2}$ $S^{-1}$) of infested plants at phenological stages V3-1; V3-2; V6; and R1 compared to their respective control conditions, non-infested plants. In the heat map, blue and orange represent up-regulated and down-regulated genes, respectively.

As for the water use efficiency (EUA), infested plants showed lower values in most of the studied cultivars, while the intrinsic rate of water use (EIUA) was similar among the cultivars (Supplementary Materials).

The photosynthetic rate (A) in infested plants was higher, while the internal $CO_2$ concentration (Ci) was proportionally lower. There was no consistent grouping pattern between cultivars regarding infested and non-infested plants, leading to different responses in the four phenological periods. The physiological parameters exhibited distinct characteristics for each cultivar in the presence of the whitefly, potentially contributing to varying degrees of resistance and/or tolerance to herbivore damage.

Cultivars Extrema, at the phenological stages V3.1, V6, and R1, and BRS9280, across all phenological stages, showed similar patterns when comparing infested and non-infested plants. Cultivars Bonus and M8808 exhibited similarities between the treatments, indicating a response to the presence of the herbivore, possibly resulting in alterations in physiological parameters (Figure 4).

**Figure 4.** Dispersion of soybean cultivars obtained by analysis of canonical variables of soybean physiological characteristics under whitefly attack in four phenological stages V3.1 (**A**), V3.2 (**B**), V6 (**C**), and R1 (**D**).

On the contrary, the GCS cultivar exhibited a unique response in the dispersion analysis between infested and non-infested plants. In contrast to the previously mentioned cultivars, GCS displayed a noticeable separation between the control and infested forms during both the V6 and R1 stages, distinguishing itself from the other cultivars. This cultivar demonstrated distinct physiological parameter behavior in plants without infestation compared to other cultivars during the V6 and R1 phenological stages (Figure 4).

Likewise, the Dominio cultivar consistently occupied opposite quadrants during the vegetative stages, signifying a notable alteration in physiological responses in the presence of the whitefly. However, the distance between the cultivars decreased relatively in the R1 stage, suggesting that the physiological parameter differences between infested and non-infested plants diminished in the later phenological stages.

For the FTR3190 cultivar, the photosynthetic parameters showed similarities between treatments in the initial phenological stages, but differed in the final phenological stages.

Regarding the cultivar M8808i, it consistently exhibited positive dispersion in both canonical variables, except for the negative dispersion observed in the R1 stage, which brought it closer to the other cultivars. This indicates a distinct physiological behavior under infestation compared to other cultivars in the vegetative phenological stage, but a similar behavior when entering the reproductive cycle.

Notably, transgenic cultivars displayed a consistent pattern of behavior in infested plants, while this behavior was not replicated when observing non-infested plants (Figure 4).

There was no grouping pattern between cultivars in terms of infested and non-infested cultivars; thus, in the four phenological periods they behaved differently, demonstrating that the physiological characters regarding the response to infestation must be different in each genotype, and thus may cause a greater or lesser degree of resistance to the insect (Figure 4).

Some cultivars showed a pattern of behaving similarly whether infested or not, and the two forms were dispersed in the same quadrant; they are Extrema in V3.1, V6 and R1; BRS9280 for all stages; Bonus that despite not being in the same quadrant whenever the control and infested are close; and M8808, always being in the same or close quadrant in the infested and control forms. These cultivars show a lack of reaction to infestation by the pest insect, implying that their physiological behavior is not altered by the presence of the whitefly.

In contrast to these cultivars, there are those that exhibit distinct physiological differences between the control and infested forms. For instance, GCS, despite sharing the same quadrant during the V6 and R1 stages, displays a complete separation between the control and infested forms, similar to the other cultivars. This indicates that the GCS cultivar displays a physiological behavior without infestation that differs from most of the other cultivars tested during the V6 and R1 stages. Another example is the Dominio cultivar, which consistently appears in opposite quadrants during vegetative stages, suggesting a substantial alteration in physiological response due to the presence of the whitefly. However, as the crop progresses, the distance between the forms decreases, becoming relatively close at the R1 stage (Figure 4).

This suggests that initially, this cultivar exhibits a heightened physiological response to infestation, which diminishes as the plant matures. In contrast to the Dominio cultivar, FTR3190 initially displays behavior similar in both the control and infested forms, but this behavior diverges over time.

It is noteworthy that Monsoy cultivars exhibit a consistent pattern of behavior when infested, a pattern that is not replicated in the control. Specifically, the cultivar M8808 consistently dispersed in the positive field for both canonical variables, except for the R1 stage, where it showed a negative dispersion and approached the behavior of the other cultivars. This indicates a physiological response under infestation that differs from other materials during the vegetative phase, but becomes more similar when entering the reproductive cycle.

## 4. Discussion

In our study, soybean plants were infested during the vegetative stage, and the nymphal density increased during the reproductive stage. This suggests that during the vegetative stage, more nutrients are available as the plant allocates photosynthate for growth, creating favorable conditions for *B. tabaci* development [13–18]. The results indicate an initial rise in photosynthetic parameters upon infestation, signifying increased energy allocation for maintaining photosynthetic functions.

The observation that resistant cultivars, demonstrating antixenosis or antibiosis against *B. tabaci*, exhibit heightened photosynthetic parameters, while susceptible cultivars consistently display reductions, implies that resistance mechanisms play a vital role in sustaining soybean plant photosynthetic efficiency under whitefly infestations. "Antixenosis" refers to characteristics deterring insect settling or feeding, while "antibiosis" negatively impacts insect growth, development, or survival when feeding on the plant [19,20].

The increase in photosynthetic parameters in resistant cultivars may be attributed to various factors. Mechanisms like reduced feeding, antixenosis, and antibiosis contribute to diminished damage to the plant's vascular system, facilitating improved nutrient and water transport, and supporting heightened photosynthetic activity. The resistance mechanisms may help mitigate stress caused by whitefly feeding, maintaining optimal physiological

conditions for photosynthesis. Resistant cultivars may better balance energy allocation for defense mechanisms and growth, contributing to higher photosynthetic efficiency. Additionally, they may produce secondary metabolites in response to whitefly feeding, some positively impacting photosynthesis and overall plant health [19,20].

In contrast, the reduction in photosynthetic parameters in susceptible cultivars could result from factors such as direct feeding damage, with greater feeding by *B. tabaci* leading to damage to the vascular system and disrupted nutrient and water transport. Susceptible plants may experience more stress due to infestation, causing alterations in hormone levels that negatively impact photosynthesis.

In healthy plants, photosynthetic parameters are typically lower compared to stressed plants [22]. *B. tabaci* feeding induces stress, evidenced by changes in soybean cultivars' photosynthetic parameters. This implies that plants depend on efficient photosynthetic performance to maintain health, and may paradoxically become more suitable hosts for *B. tabaci*.

Whiteflies are phloem-feeders, and phloem transports photoassimilates, including sugars [23]. Sucrose and glucose are primary sugars associated with high occurrences of whiteflies, promoting their survival, adult longevity, fecundity, feeding, oviposition preference, and immature development [24]. While these sugars are closely linked to photosynthesis, our study observed a positive relationship between *B. tabaci* infestation and photosystem activity.

Quantifying *B. tabaci* damage presents challenges due to indirect effects on productivity and potential confounding impacts from other pests during the soybean production cycle. Evaluating *B. tabaci* effects on photosynthesis can serve as an alternative to indirectly assess damage. Previous studies reported photosynthetic rate reductions caused by *B. tabaci* infestations in various crops [16,25,26].

Our findings indicate that *B. tabaci* feeding affects plant physiology, reflected in relationships among photosynthetic parameters and levels of sugars and starch. The most significant impacts occurred during the vegetative stage, when plants were more suitable for *B. tabaci* development, inducing stress with potential consequences on productivity [15,27].

In plants infested with herbivores, a decline in the photosynthetic rate and alterations in variables linked to photosynthesis may take place. Our study noted an elevation in the photosynthetic rate in cultivars M8808 and M8349 towards the conclusion of the V6 growth stage, potentially linked to $CO_2$ assimilation via rubisco, suggesting that the augmented rate in these cultivated plants is linked to the $CO_2$ assimilation phase.

Schutze et al. (2022) [18] observed higher chlorophyll content in the vegetative stage for all levels of *B. tabaci* infestation. In our study, where plants were infested at the beginning of the vegetative period and remained infested throughout the phenological cycle, we also noted an increase in photosynthetic parameters in infested plants. Yee et al. (1996) [27] found similar results, reporting higher photosynthetic rates and stomatal conductance in cotton plants infested with *B. argentifolii*. These findings indicate that *B. tabaci* feeding significantly impacts plant physiology, particularly related to photosynthetic parameters.

Overall, our study highlights the importance of considering the timing of insect infestation, as the greatest impacts on plant physiology occurred during the vegetative phenological stage. This information is valuable for understanding *B. tabaci* effects on soybean plants, with implications for productivity. The findings underscore the importance of plant resistance in maintaining photosynthetic activity and overall plant health when facing insect infestations, providing valuable insights for breeding and selecting resilient soybean cultivars, ultimately contributing to more sustainable and productive agricultural practices.

**Supplementary Materials:** The following supporting information can be downloaded at: https://www.mdpi.com/article/10.3390/agronomy14030481/s1. ANOVA breakdown, for all photosynthetic parameters.

**Author Contributions:** Conceptualization, L.B.S., R.d.S.M. and B.E.P.; data curation, B.E.P., L.d.S.F., L.C.A., C.M.P.d.S., G.d.S.R., J.V.S.M., A.F.T.d.S., H.M. and N.S.d.R.; formal analysis, L.C.A., R.R.d.B., R.d.S.M., V.M.G.d.C.L., T.R.d.O., R.H.F.R., M.C.F.e.S., L.B.S. and H.M.; funding acquisition, L.B.S. and R.d.S.M.; investigation, L.C.A., M.C.F.e.S., T.R.d.O. and R.H.F.R.; methodology, J.d.J.L., B.E.P., L.d.S.F., L.C.A., C.M.P.d.S., G.d.S.R., J.V.S.M., A.F.T.d.S., H.M. and N.S.d.R.; resources and software, B.E.P., R.R.d.B., L.B.S., R.H.F.R., V.M.G.d.C.L., N.S.d.R. and L.d.S.F.; supervision, L.B.S., R.R.d.B., R.d.S.M. and L.d.S.F.; visualization, B.E.P., L.d.S.F., L.C.A., C.M.P.d.S., G.d.S.R., J.V.S.M., A.F.T.d.S., H.M. and N.S.d.R.; writing—original draft, B.E.P., R.R.d.B., L.B.S., R.H.F.R., V.M.G.d.C.L., N.S.d.R. and L.d.S.F.; writing—review and editing, B.E.P., J.d.J.L., R.R.d.B., L.B.S., R.H.F.R., V.M.G.d.C.L., N.S.d.R., G.d.S.R., J.V.S.M. and T.R.d.O. All authors have read and agreed to the published version of the manuscript.

**Funding:** National Council for Scientific and Technological Development—CNPq–004/2020; Coordination for the Improvement of Higher Education Personnel (CAPES); scholarship from Piauí State Research Foundation (FAPEPI) 008/2018.

**Data Availability Statement:** Data are contained within the article.

**Acknowledgments:** We thank the National Council for Scientific and Technological Development—CNPq—Brazil for financial support. We also thank the Coordination for the Improvement of Higher Education Personnel (CAPES), Brazil, the Piauí State Research Foundation (FAPEPI) for the scholarships and resources provided, and the Federal University of Piauí, Brazil, for providing logistical support.

**Conflicts of Interest:** The authors declare no conflict of interest.

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
