# Peer review of "Change in the Physiological Aspects of Soybean Caused by Infestation by Bemisia tabaci MEAM1"

_agronomy, doi:10.3390/agronomy14030481_

Round 1
Reviewer 1 Report
Comments and Suggestions for Authors
Silva et al. reported the physiological aspects of soyabean caused by infestation by B. tabaci, here are my comments:
Ref. 18 reported "The goal was to identify the optimal parameter to directly quantify pest damage on crop yield. Correlation networks were created among data on sugar content (fructose, glucose, and sucrose), starch and photosynthetic parameters (initial fluorescence, performance index on absorption basis, and turn-over number), and the number of nymphs at each of three infestations level (low, medium, and high) during both the vegetative and reproductive stage of the crop." while the author of the current study reported "The evaluation of photosynthetic parameters, such as photo- 21 synthetic rate, leaf transpiration, stomatal conductance, and internal CO2 concentration, revealed 22 that B. tabaci infestation influenced gas exchange in soybean plants. How the current study is different/novel from the previous? both studies are conducted on soyabean with B. tabaci infestation.
Ref 12 reported, " we measured the plant growth and chlorophyll content of tobacco and cotton plants that were infested by MEAM1 nymphs. Furthermore, to investigate the spatial and temporal changes in photosynthesis caused by MEAM1 nymphs, the net photosynthetic rate (Pn) and chlorophyll a fluorescence of local and systemic tobacco leaves were assayed at 8, 11, 14, and 20 days after MEAM1 adult removal, which represent the stages of 1st, 2nd, 3rd, and 4th instar nymphs, respectively." while Ref 13 reported. "Tomato plants ‘Santa Adélia Super’ infested with B. tabaci (MED and MEAM1), and non-infested plants were evaluated for differences in gas exchange, chlorophyll - a fluorescence of photosystem II (PSII), and biochemical factors (total phenols, total flavonoids, superoxide dismutase—SOD, peroxidase—POD, and polyphenol oxidase—PPO)." Please state how the current study is different/novel from the above?
L 58, "Given the significant impact of B. tabaci MEAM1 on soybean crops, the objective of 58 this study was to assess the physiological characteristics of susceptible and resistant soy- 59 bean cultivars to B. tabaci." but I think these physiological characteristics has been studied before, isn't?
L 360: "Overall, our study highlights the importance of considering the timing of insect infestation, as the greatest impacts on plant physiology occurred during the vegetative phenological stage." while Ref 18 reported in conclusion "From our results, the greatest impacts of insect infestation occur during the vegetative phenological stage," Isn't both studies are the same?
Please state clearly the rationale why the current study is important? How it different from the ref. 18? how the current results are same/different from the Ref. 18?
other typo, please italic name of B. tabaci in MS
Author Response
Dear reviewer We appreciate the analysis carried out on the manuscript, we believe we have answered all the questions and corrected the errors detected. In Ref 18, the authors address the damage associated with the density of individuals in a soybean cultivar. In the present study, an evaluation was carried out on 11 cultivars, to observe the profile of the herbivory process, associated with gas exchange.
In Refs 12 and 13,
we have important contributions to the understanding of insect-plant interaction, with regard to gas exchange and plant defense mechanisms. The present study contributes to the understanding of the insect-plant interaction - soybean and Bemisia tabaci, regarding the response of soybean cultivars, during herbivory by B. tabaci, not regarding photosynthetic processes. Find out whether changes in photosynthetic processes can interfere with plant performance. Yes, the cultivars investigated in the present study have already been investigated by the group, as can be seen in the references: 19. Rodrigues, RHF; Silva, L.B.; Almeida, LFO; Silva, S.R. Yes; Sobrinho, N.; Maggioni, K.; Rodrigues, TF; and Pavan, B. E. 2021. “Vertical Distribution of Bemisia tabaci in Soybean and Cultivar Attractiveness.” Entomologia Experimentalis et Applicata, April, eea.13051. https://doi.org/10.1111/eea.13051. 20. Silva, MCF; Arielly, S.; Rodrigues, RHF; Pavan, BE; Silva, L. B. 2023. Performance of Bemisia tabaci MEAM1 in soybean and resistance characteristics of cultivars. Asia-Pacific Journal of Entomology, 26, 2, 1226-8615. https://doi.org/10.1016/j.aspen.2023.102100. All B. tabci names have been italicized.
the present study was supported for the development of the work by ref. 18, which corroborates the data obtained. The present study makes an important contribution to studies on improving soybean crops, and with continued research, we will be able to make a correlation between cultivars resistant or susceptible to herbivory by B. tabaci. Here we report the behavior of 11 cultivars. This data, added to other articles published by the group, will support the soybean breeding industry, as well as production analysis.

Reviewer 2 Report
Comments and Suggestions for Authors
Please find the comments and suggestions in the attached document.

The paper is well-written in Abstract and Introduction. But English needs to be improved in Materials and methods, Results, and Discussion.
Author Response
Dear Reviewer, We inform you that we seek to respond to all suggestions for corrections and questions regarding the manuscript. the questions were answered in the PDF file and the coections were made in the word manuscript.
https://drive.google.com/file/d/1iC1mMIWdetcuoZnMljsLXK0yNZLQd1lS/view?usp=drive_link

Round 2
Reviewer 1 Report
Comments and Suggestions for Authors
As the author stated that "we believe we have answered all the questions and corrected the errors detected." but I am still confuse as there is no cover letter or point to point response. please mention in a point by point and with line number in a cover letter or reply to the reviewers.
please mention to the reader in introduction and discussion where necessary the rationale and difference of this study from the previous study(ies)...
Author Response
Reply letter
Dear.
We appreciate the contributions made by the reviewers; the vast majority of suggested changes were made.
Reviewer:.
Editor and Reviewer comments:    
The revision process respected most of the review comments and improved the quality of manuscript. However, a few minor comments should be addressed.
- 18 reported "The goal was to identify the optimal parameter to directly quantify pest damage on crop yield. Correlation networks were created among data on sugar content (fructose, glucose, and sucrose), starch and photosynthetic parameters (initial fluorescence, performance index on absorption basis, and turn-over number), and the number of nymphs at each of three infestations level (low, medium, and high) during both the vegetative and reproductive stage of the crop." while the author of the current study reported "The evaluation of photosynthetic parameters, such as photo- 21 synthetic rate, leaf transpiration, stomatal conductance, and internal CO2 concentration, revealed that B. tabaci infestation influenced gas exchange in soybean plants. How the current study is different/novel from the previous? both studies are conducted on soyabean with B. tabaci infestation.
Answer: The main objective of the present study is to investigate soybean cultivars resistant to B. tabaci. The group has been investigating cultivars, in order to seek data that can feed the soybean improvement industry, regarding the main insects of economic importance. Thus, we investigated the behavior of B. tabaci in soybean cultivars and in this study the plant's response to B. tabaci herbivory, purchasing resistant and susceptible soybean cultivars to B. tabaci outlined in previous studies.
2 - Ref 12 reported, " we measured the plant growth and chlorophyll content of tobacco and cotton plants that were infested by MEAM1 nymphs. Furthermore, to investigate the spatial and temporal changes in photosynthesis caused by MEAM1 nymphs, the net photosynthetic rate (Pn) and chlorophyll a fluorescence of local and systemic tobacco leaves were assayed at 8, 11, 14, and 20 days after MEAM1 adult removal, which represent the stages of 1st, 2nd, 3rd, and 4th instar nymphs, respectively." while Ref 13 reported. "Tomato plants ‘Santa Adélia Super’ infested with B. tabaci (MED and MEAM1), and non-infested plants were evaluated for differences in gas exchange, chlorophyll - a fluorescence of photosystem II (PSII), and biochemical factors (total phenols, total flavonoids, superoxide dismutase—SOD, peroxidase—POD, and polyphenol oxidase—PPO)." Please state how the current study is different/novel from the above?
Answer: The current study differs from those reported in the literature and in the present study, in that it evaluates soybean cultivars resistant and susceptible to B. tabaci, proven in previous studies of these cultivars. in the production of data for the genetic improvement of soybean cultivars
3 - L 58, "Given the significant impact of B. tabaci MEAM1 on soybean crops, the objective of 58 this study was to assess the physiological characteristics of susceptible and resistant soy- 59 bean cultivars to B. tabaci." but I think these physiological characteristics has been studied before, isn't?
 
Answer: Yes, there have been studies before, focusing on evaluating the plant's response to B. tabaci herbivory, the present study seeks to compare resistant and susceptible cultivars - response of soybean cultivars to B. tabaci herbivory
4 - L 360: "Overall, our study highlights the importance of considering the timing of insect infestation, as the greatest impacts on plant physiology occurred during the vegetative phenological stage." while Ref 18 reported in conclusion "From our results, the greatest impacts of insect infestation occur during the vegetative phenological stage," Isn't both studies are the same?
Answer: Yes, the two studies proved that there is no phenological study showing the greatest impact of B. tabaci herbivory on soybean photosynthetic processes, under hebivory. However, in the present study the comparison of resistant and susceptible cultivars to B. tabaci - these data will contribute to the improvement of soybean cultivars
5 - Please state clearly the rationale why the current study is important? How it different from the ref. 18? how the current results are same/different from the Ref. 18?
Answer: The present study differs from those already conducted with soybean plants under herbivory, in that we are evaluating different soybean cultivars with different behaviors against herbivory. We seek to analyze this behavior by purchasing resistant and susceptible cultivars, to increase the production industry's database. of soybean cultivars that are more resistant to insects, diseases and have greater productivity.
6 - other typo, please italic name of B. tabaci in MS
Answer: The name of B. tabaci has been corrected throughout the body of the work
Modifications made to the manuscript
1 - The abstract has undergone revision.
2 - The introduction has been enhanced with the inclusion of information pertaining to biotypes B.
3 - Regarding the editor's comment in the conclusion, it should be noted that the existing literature does not provide definitive evidence. However, we would like to inform you that we are currently engaged in an analysis of the metabolome of these cultivars. This analysis aims to unravel the resistance factors that contribute to reduced oviposition in specific cultivars.
* All changes made to the manuscript are highlighted.